# ROBUST TRANSFER LEARNING BASED ON MINIMAX PRINCIPLE

## ABSTRACT

The similarity between target and source tasks is a crucial quantity for theoretical analyses and algorithm designs in transfer learning studies. However, this quantity is often difficult to be precisely captured. To address this issue, we make a boundedness assumption on the task similarity and then propose a mathematical framework based on the minimax principle, which minimizes the worst-case expected population risk under this assumption. Furthermore, our proposed minimax problem can be solved analytically, which provides a guideline for designing robust transfer learning models. According to the analytical expression, we interpret the influences of sample sizes, task distances, and the model dimensionality in knowledge transferring. Then, practical algorithms are developed based on the theoretical results. Finally, experiments conducted on image classification tasks show that our approaches can achieve robust and competitive accuracies under random selections of training sets.

## 1 INTRODUCTION

The goal of the transfer learning is to solve target tasks by the learning results from some source tasks. In order to study the fundamental aspects of the transfer learning problems, it is important to define and quantify the similarity between source and target tasks (Pan & Yang, 2009). While it is assumed that the source and target tasks are kind of similar in transfer learning problems (Weiss et al., 2016), the joint structures and similarity between the tasks can only be learned from the training data, which is challenging to be practically computed due to the limited availability of the labeled target samples. Therefore, in order to conduct meaningful theoretical analyses, it is often necessary to make extra assumptions, such as the linear combination of learning results (Ben-David et al., 2010) and linear regression transferring (Kuzborskij & Orabona, 2013), which could be limited in many applications.

As such, in this paper, we attempt to theoretically study the transfer learning by only assuming that the similarity between the source and target tasks is bounded, which is a weaker assumption, and is often valid in transfer learning problems. Under such an assumption, the *minimax principle* can be applied (Verdu & Poor, 1984) for estimating the target distribution. Based on this principle, the estimator minimizes the worst-case *expected population risk* (EPR) (Jin et al., 2018) under the bounded task distance constraint, which maintains robustness against the weak assumption. Practically, many empirical works have also followed the minimax setting and verify its validness (Zhang et al., 2019), while the theoretical analyses appear to be rather behind. The main challenge of analyzing general minimax problems in transfer learning is due to the difficulty of computing the expectations of the population risk under popular distance measures, such as the Kullback–Leibler (K-L) divergence (Thomas & Joy, 2006).

To deal with this difficulty, we adopt the widely used $\chi^2$-distance and Hellinger distance (Csiszár & Shields, 2004) as the distance measure between data distributions of the tasks, and present a minimax formulation of transfer learning. By adopting such measures, the proposed minimax problems can be analytically solved. In particular, we show that the optimal estimation is to linearly combine the learning results of two tasks, where the combining coefficient can be computed from the training data. This provides a theoretical justification for many existing analyzing framework and algorithms (Ben-David et al., 2010; Garcke & Vanck, 2014). Note that the recent work (Tong et al., 2021) also analytically evaluates the combining coefficients, which rely on the underlying task distributions that are not available for real applications. Our work essentially provides the combining coefficient that

are both theoretical optimal and computable from data, which can be more appealing in practical applications.

Moreover, the analyses of the minimax transfer learning problem on discrete data can be extended to the continuous data for real applications. In the continuous case, we consider similar transfer learning scheme as in (Nguyen et al., 2020), which transfers the topmost layer of the neural networks between source and target tasks. In particular, we show the analytical solution of optimal weights in the topmost layer, which is again a linear combination of the weights of the source and target problem. Furthermore, we propose the transfer learning algorithm guided by the theoretical results, where the robustness and performances of the algorithms are validated by several experiments on real datasets.

The contribution of this paper can be summarized as follows:

- We make mild assumptions of the task distance and propose a minimax framework for analyzing transfer learning. Additionally, we establish the analytical solutions of the minimax problems in the discrete data space.

- We extend the analyses to continuous data and establish similar results for the learning models with neural networks. Furthermore, we apply our theoretical results to develop robust transfer learning algorithms.

- The experiments in real datasets validate our proposed algorithms, where our approaches can have higher robustness and competitive accuracy.

Due to the space limitations, the proofs of theorems are presented in the supplemental materials.

## 2 PROBLEM FORMULATION

### 2.1 NOTATIONS AND DEFINITIONS

We denote $X$ and $Y$ as the random variables of data and label with domains $\mathcal{X}$ and $\mathcal{Y}$, respectively. For ease of illustration, the data $X$ is set as a discrete random variable in section 2 and section 3.

We consider the transfer learning problem that has a target task and a source task, denoted as task $T$ and $S$, respectively. For each task $i = T, S$, there are $n_i$ training samples $\{(x_\ell^{(i)}, y_\ell^{(i)})\}_{\ell=1}^{n_i}$ i.i.d. generated from the underlying joint distributions $P_{XY}^{(i)}$ with[1] $P_{XY}^{(i)}(x, y) > 0$, for all $(x, y) \in \mathcal{X} \times \mathcal{Y}$. The empirical distributions $\hat{P}_{XY}^{(i)}$ $(i = T, S)$ of the samples are defined as

$$\hat{P}_{XY}^{(i)}(x, y) \triangleq \frac{1}{n_i} \sum_{\ell=1}^{n_i} \mathbb{1}\{x_\ell^{(i)} = x, y_\ell^{(i)} = y\},$$

where $\mathbb{1}\{\cdot\}$ denotes the indicator function (Feller, 2008) and let $\mathcal{P}_n$ be the set of all the possible empirical distributions supported by $\mathcal{X} \times \mathcal{Y}$ with $n$ samples.

In this paper, we employ the following two distance measures for probability distributions, which are also widely used in statistics (Csiszár & Shields, 2004), and more convenient in our analyses.

**Definition 2.1** (Referenced $\chi^2$-distance). *Let $R(z)$, $P(z)$, and $Q(z)$ be the distributions supported by $\mathcal{Z}$. The $\chi^2$-distance between $P(z)$ and $Q(z)$ referenced by $R(z)$ is defined as follows,*

$$\chi_R^2(P, Q) \triangleq \sum_{z \in \mathcal{Z}} \frac{(P(z) - Q(z))^2}{R(z)}. \tag{1}$$

**Definition 2.2** (Hellinger Distance). *Let $P(z)$ and $Q(z)$ be the distributions supported by $\mathcal{Z}$, The Hellinger distance between $P(z)$ and $Q(z)$ is defined as follows.*

$$H^2(P, Q) \triangleq \frac{1}{2} \sum_{z \in \mathcal{Z}} \left( \sqrt{P(z)} - \sqrt{Q(z)} \right)^2, \ and \ H(P, Q) \triangleq \sqrt{H^2(P, Q)}. \tag{2}$$

---

[1]This assumption comes from the fact that in practice such joint distributions are typically modeled by some positive parameterized families, e.g., the softmax function.

## 2.2 MINIMAX FORMULATION

Since estimating the similarity between target and source tasks from data is challenging, we attempt to only make the assumption that the distance between two tasks is bounded by some constant $D$ under a distance measure $d(\cdot, \cdot)$, i.e.,

$$d\left(P_{XY}^{(T)}, P_{XY}^{(S)}\right) \leq D. \tag{3}$$

Based on this assumption, we consider a minimax formulation for estimating the target distribution with estimator $Q_{XY}(\hat{P}_{XY}^{(T)}, \hat{P}_{XY}^{(S)})$, where the notation represents that the estimator $Q_{XY}$ is a function of the empirical distributions $\hat{P}_{XY}^{(T)}$ and $\hat{P}_{XY}^{(S)}$.

**Minimax Formulation:**

$$\min_{Q_{XY}(\hat{P}_{XY}^{(T)}, \hat{P}_{XY}^{(S)})} \max_{P_{XY}^{(S)}: d(P_{XY}^{(T)}, P_{XY}^{(S)}) \leq D} \mathbb{E}\left[d\left(P_{XY}^{(T)}, Q_{XY}\right)\right], \tag{4}$$

where the expectation is taken over all possible $\hat{P}_{XY}^{(T)}$ and $\hat{P}_{XY}^{(S)}$ in $\mathcal{P}_{n_T}$ and $\mathcal{P}_{n_S}$.

This formulation can be divided into two parts: (1) for given estimator $Q_{XY}$, we consider the largest expected population risk (EPR) of the distance between the underlying target distribution and the estimator; (2) we find the best estimator $Q_{XY}$ as the function of training data that could minimize the worst risk. Note that the empirical distributions are sufficient statistics for the underlying distributions (Van der Vaart, 2000). We therefore consider $Q_{XY}$ as the function of both empirical distributions. Accordingly, the EPR of the derived estimator under the true similarity is always smaller than the result of formulation (4). In other word, we design an estimator that has an upper-bounded EPR and it thus leads to robustness.

Notice that the formulation (4) is generally difficult to be solved analytically due to: (i) the distance measure $d(\cdot, \cdot)$ can cause difficulty in computation, e.g., the logarithm function in the K-L divergence; (ii) the expectations over $\hat{P}_{XY}^{(T)}$ and $\hat{P}_{XY}^{(S)}$ follow the multinomial distribution (Csiszár, 1998), i.e., the probability of the empirical distribution $\mathbb{P}(\hat{P}_{XY}^{(i)}; P_{XY}^{(i)}) \propto \exp(-n_i D(\hat{P}_{XY}^{(i)} \| P_{XY}^{(i)}))$, which is complicated to analyze. To address the issue (i), we choose the $\chi^2$-distance and Hellinger distance, which are more convenient to be analyzed in minimax problems. Moreover, for the issue (ii), we propose to study the surrogate problem which replaces the expectation computation in (4) by the integral

$$\iint d\left(P_{XY}^{(T)}, Q_{XY}\right) \prod_{i=T,S} \exp\left(-n_i d\left(\hat{P}_{XY}^{(i)}, P_{XY}^{(i)}\right)\right) d\hat{P}_{XY}^{(i)}. \tag{5}$$

Note that (5) is the asymptotic approximation of the expectation over multinomial distributions with the additional surrogation that the exponent in (5) can be chosen different from the K-L divergence. Such asymptotic approximation is also applied for theoretical analyses in high dimensional statistics (Morris, 1975). Then, the goal of this paper is to study the following minimax problems for transfer learning.

**Formulation 1 (referenced $\chi^2$-distance):**

$$\min_{Q_{XY}(\hat{P}_{XY}^{(T)}, \hat{P}_{XY}^{(S)})} \max_{P_{XY}^{(S)}: \chi_R^2(P_{XY}^{(T)}, P_{XY}^{(S)}) \leq D^2} \mathbb{E}\left[\chi_R^2(P_{XY}^{(T)}, Q_{XY})\right], \tag{6}$$

where the expectation is the integral over

$$\mathbb{P}\left(\hat{P}_{XY}^{(i)}; P_{XY}^{(i)}\right) \propto \exp\left(-\frac{n_i}{2} \chi_R^2\left(\hat{P}_{XY}^{(i)}, P_{XY}^{(i)}\right)\right), \ i = T, S. \tag{7}$$

Note that the referenced-$\chi^2$ distance can be recognized as an asymptotic approximation of K-L divergence by Lemma A.1 in Appendix A. Here the reference distribution $R$ is selected as $\hat{P}_{XY}^{(S)}$.[2]

**Formulation 2 (Hellinger distance):**

$$\min_{Q_{XY}(\hat{P}_{XY}^{(T)}, \hat{P}_{XY}^{(S)})} \max_{P_{XY}^{(S)}: H(P_{XY}^{(T)}, P_{XY}^{(S)}) \leq D} \mathbb{E}\left[H^2\left(P_{XY}^{(T)}, Q_{XY}\right)\right], \tag{8}$$

---

[2]Since the source samples are sufficient, with high probability, all the entries of $\hat{P}_{XY}^{(S)}$ are positive.

where the expectation is the integral over

$$\mathbb{P}\left(\hat{P}_{XY}^{(i)}; P_{XY}^{(i)}\right) \propto \exp\left(-2n_i H^2\left(\hat{P}_{XY}^{(i)}, P_{XY}^{(i)}\right)\right), \ i = T, S. \tag{9}$$

Note that Hellinger distance provides a lower bound of the K-L divergence with Lemma A.2 in Appendix A, and thus Formulation 2 computes a lower bound of the population risk in (4).

## 3 ANALYSES FOR DISCRETE DATA

In this section, we provide the analytical solutions of the formulations (6) and (8). Similar minimax estimation problems have been studied in early works (Trybula, 1958; Berry, 1990). We now directly give the detailed expressions and the proof is provided in the supplementary material.

### 3.1 ANALYTICAL SOLUTION OF FORMULATION 1

**Theorem 3.1.** *Let $Q_{XY}^{(1)}$ be the estimator that achieves the minimax solution of problem (6) and then[3]*

$$Q_{XY}^{(1)}(x, y) = (1 - \alpha_1)\hat{P}_{XY}^{(T)}(x, y) + \alpha_1 \hat{P}_{XY}^{(S)}(x, y), \tag{10}$$

*for all $(x, y) \in \mathcal{X} \times \mathcal{Y}$, where*

$$\alpha_1 \triangleq \frac{n_S}{n_T + n_S}\left(1 - \frac{I_{\frac{|\mathcal{X}||\mathcal{Y}|}{2}}\left(\frac{n_T n_S DD_1}{n_T + n_S}\right)}{I_{\frac{|\mathcal{X}||\mathcal{Y}|}{2}-1}\left(\frac{n_T n_S DD_1}{n_T + n_S}\right)}\frac{D}{D_1}\right), \tag{11}$$

*and $D_1 \triangleq \sqrt{\chi_R^2(\hat{P}_{XY}^{(T)}, \hat{P}_{XY}^{(S)})}$. Specifically, $I_\nu(\cdot)$ denotes the modified Bessel function of the first kind with order $\nu$ (Abramowitz et al., 1988), whose definition is in Appendix B.*

In the remaining parts of this paper, we denote for ease of presentation $J_\nu(x) \triangleq I_{\frac{\nu}{2}}(x)/I_{\frac{\nu}{2}-1}(x)$.

Theorem 3.1 implies that linearly combining the learning results of different tasks is a preferable method for robust transfer learning, which is also widely used in existing algorithms and theoretical frameworks (Ben-David et al., 2010). Moreover, these works intuitively assume it, whereas we provide a theoretic support that could help explain the rationality.

**Remark 3.2.** *To help understand the related factors contained in (11), we consider a special regime that $\nu \gg x$, where $J_\nu(x) \sim x/\nu$. Then the expression of (11) can be approximated by*

$$\alpha_1 \sim \frac{n_S}{n_T + n_S}\left(1 - \frac{n_T n_S D^2}{(n_S + n_T)|\mathcal{X}||\mathcal{Y}|}\right). \tag{12}$$

*This result is consistent with Eq.(6) in (Tong et al., 2021) under this special regime as explained in Appendix C. The coefficient $\alpha_1$, which represents the requirement of source samples, is positively associated with the model dimensionality $|\mathcal{X}||\mathcal{Y}|$, which comes from that we learn all the $|\mathcal{X}||\mathcal{Y}|$ entries of the target distribution, and negatively associated with the target sample size $n_T$ and task distance $D$. These relationships are examined in the experimental part. We also provide an interesting geometric explanation of this pattern, which is shown in Appendix C.*

### 3.2 ANALYTICAL SOLUTION OF FORMULATION 2

In the following, we provide the solution of problem (8) based on Hellinger distance.

**Theorem 3.3.** *Let $Q_{XY}^{(2)}$ be the estimator that achieves the minimax solution of problem (8) and then[4]*

$$Q_{XY}^{(2)}(x, y) = \left[(1 - \alpha_2)\sqrt{\hat{P}_{XY}^{(T)}(x, y)} + \alpha_2\sqrt{\hat{P}_{XY}^{(S)}(x, y)}\right]^2,$$

*for all $(x, y) \in \mathcal{X} \times \mathcal{Y}$, where*

$$\alpha_2 = \frac{n_S}{n_T + n_S}\left(1 - \frac{D}{D_2}J_{|\mathcal{X}||\mathcal{Y}|}\left(\frac{4n_S n_T DD_2}{n_T + n_S}\right)\right),$$

*and $D_2 \triangleq H(\hat{P}_{XY}^{(T)}, \hat{P}_{XY}^{(S)})$.*

Accordingly, we can also achieve a similar interpretation of the affecting factors as in section 3.1.

---

[3]This solution actually requests $D/\sqrt{1/n_T + 1/n_S} \leq \sqrt{|\mathcal{X}||\mathcal{Y}|}$, which can be easily guaranteed.
[4]It requests $D/\sqrt{1/4n_T + 1/4n_S} \leq \sqrt{|\mathcal{X}||\mathcal{Y}|}$.

# 4 CONTINUOUS CASE AND ALGORITHM

In this section, we extend the previous analyses of discrete data to continuous data, which conform to the practical setting. In such cases, the previously adopted empirical distributions can not be seen as valid observations due to the infinite cardinality $|\mathcal{X}|$, where most of the possible data are not sampled.

In order to apply the previous analyzing framework, we consider the retrain-head method (Nguyen et al., 2020), which is a commonly used transfer learning technique. With this method, a pre-trained network is prepared for extracting the feature of the data and then the topmost layer (known as "*head*") can be retrained with observed samples. Under such a setting, we can recognize the weights in retrained topmost layers as the observations from the corresponding tasks. Note that compared with the cardinality of the large data space, the topmost layer has much fewer parameters.

In detail, a pre-trained network is composed of two parts of networks: (a) the previous layers whose input is data $x$ and output is the $d$-dimensional features $\boldsymbol{f}(x) \in \mathbb{R}^d$, and (b) the topmost layer for linear classification, with weights $\boldsymbol{g}(y) \in \mathbb{R}^d$ of each label $y$. Each task provides the learned weights to estimate the optimal weights for the target task. Moreover, the learned weights are decided by the model proposed for describing the data. For theoretical analyses, we provide the discriminative models for $\chi^2$-distance and Hellinger distance, where thereupon we can design practical algorithms. For convenience, we use notation $\boldsymbol{h}$ to represent the topmost layer in section 4.2.

## 4.1 REVISED FORMULATION 1 AND MM-$\chi^2$ ALGORITHM

When $\chi^2$-distance is chosen as the distance measure, we consider the discriminative model for the target distribution in the factorization form

$$Q_{Y|X}^{(\boldsymbol{f},\boldsymbol{g})}(y|x) \triangleq P_Y^{(T)}(y)\left(1 + \boldsymbol{f}^{\mathrm{T}}(x)\boldsymbol{g}(y)\right), \tag{13}$$

which provides the probability of each label $y \in \mathcal{Y}$ for any data $x$. Such a model has been introduced in factorization machines (Rendle, 2010) and is commonly used in natural language processing problems (Levy & Goldberg, 2014).

Under the pre-trained feature extractor $\boldsymbol{f}^*(\cdot)$, the learned weights of topmost layers can be derived by minimizing the distance between the empirical distribution and the model. For computation, we avoid using the joint distribution as the reference and define the $\chi^2$-distance measure referenced by the product marginal distribution, i.e., $\chi_M^2(\cdot,\cdot) \triangleq \chi_{P_X^{(T)}P_Y^{(T)}}^2(\cdot,\cdot)$. Then, the learned weights $\hat{\boldsymbol{g}}_i$ of each task $i = T, S$ can be defined as

$$\hat{\boldsymbol{g}}_i \triangleq \arg\min_{\boldsymbol{g}} \chi_M^2\left(\hat{P}_{XY}^{(i)}, P_X^{(T)}Q_{Y|X}^{(\boldsymbol{f}^*,\boldsymbol{g})}\right). \tag{14}$$

Now $\hat{\boldsymbol{g}}_T$ and $\hat{\boldsymbol{g}}_S$ are the observations to generate the minimax solution, where the expectations can be defined as $\boldsymbol{g}_i(y) = \mathbb{E}[\hat{\boldsymbol{g}}_i(y)]$. Note that the parameters of $\boldsymbol{g}_T$ are just the topmost weights we hope to achieve. Then, the minimax problem can be defined as follows [cf. (6)]:

$$\boldsymbol{g}^* = \arg\min_{\boldsymbol{g}(\hat{\boldsymbol{g}}_T,\hat{\boldsymbol{g}}_S)} \max_{\boldsymbol{g}_S \in \mathcal{G}} \mathbb{E}\left[\chi_M^2(P_X^{(T)}Q_{Y|X}^{(\boldsymbol{f}^*,\boldsymbol{g}_T)}, P_X^{(T)}Q_{Y|X}^{(\boldsymbol{f}^*,\boldsymbol{g})})\right], \tag{15}$$

where $\mathcal{G} \triangleq \left\{\boldsymbol{g} : \chi_M^2(P_X^{(T)}Q_{Y|X}^{(\boldsymbol{f}^*,\boldsymbol{g}_T)}, P_X^{(T)}Q_{Y|X}^{(\boldsymbol{f}^*,\boldsymbol{g})}) \leq D^2\right\}$.

We can directly apply Theorem 3.1 and obtain the following theorem.

**Theorem 4.1.** *When the empirical distributions $\hat{P}_{XY}^{(T)}$ and $\hat{P}_{XY}^{(S)}$ follow the density function (7), the minimax solution as defined in (15) is* [5]

$$\boldsymbol{g}^* = (1 - \tilde{\alpha}_1)\hat{\boldsymbol{g}}_T + \tilde{\alpha}_1 \hat{\boldsymbol{g}}_S, \tag{16}$$

*where*

$$\tilde{\alpha}_1 = \frac{n_S}{n_T + n_S}\left(1 - \frac{D}{\tilde{D}_1}J_{d|\mathcal{Y}|}\left(\frac{n_T n_S D \tilde{D}_1}{n_T + n_S}\right)\right).$$

*and $\tilde{D}_1^2 \triangleq \chi_M^2(P_X^{(T)}Q_{Y|X}^{(\boldsymbol{f}^*,\hat{\boldsymbol{g}}_T)}, P_X^{(T)}Q_{Y|X}^{(\boldsymbol{f}^*,\hat{\boldsymbol{g}}_S)})$.*

---

[5]In practice, when the assumption $D/\sqrt{1/n_T + 1/n_S} < \sqrt{d|\mathcal{Y}|}$ does not hold, the estimator can still provide a sub-optimal solution.

---

**Algorithm 1** Minimax $\chi^2$-Algorithm (MM-$\chi^2$)

---

1: **Input:** target and source data samples $\{(x_l^{(i)}, y_l^{(i)})\}_{l=1}^{n_i}$ $(i = T, S)$, learning rate $\eta$
2: Randomly initialize $\alpha$, $\boldsymbol{f}^*$, $\boldsymbol{g}^*$
3: **repeat**
4: $\quad (\boldsymbol{f}^*, \boldsymbol{g}^*) \leftarrow (\boldsymbol{f}^*, \boldsymbol{g}^*) - \eta \nabla_{(\boldsymbol{f}, \boldsymbol{g})} L_1(\alpha, \boldsymbol{f}^*, \boldsymbol{g}^*)$
5: $\quad \alpha \leftarrow \frac{n_S}{n_T + n_S} \left( 1 - \frac{D}{D_1} J_{d|\mathcal{Y}|} \left( \frac{n_T n_S D \tilde{D}_1}{n_T + n_S} \right) \right)$
6: **until** $\boldsymbol{f}^*$, $\boldsymbol{g}^*$ converge
7: **return** $\boldsymbol{f}^*, \boldsymbol{g}^*$

---

Accordingly, we can design an algorithm based on Theorem 4.1. Despite the theoretical analyses where the feature extractor is fixed, our algorithm jointly optimizes the feature extraction $\boldsymbol{f}$ and the topmost layers $\boldsymbol{g}$, which is a typical retraining procedure. It is proved that the linearly combined weights (16) can be achieved by minimizing the linearly combined training loss with the same coefficient. We therefore define

$$L_1(\alpha, \boldsymbol{f}, \boldsymbol{g}) \triangleq (1 - \alpha)\chi_M^2 \left( \hat{P}_{XY}^{(S)}, P_X^{(T)} Q_{Y|X}^{(\boldsymbol{f}, \boldsymbol{g})} \right) + \alpha \chi_M^2 \left( \hat{P}_{XY}^{(T)}, P_X^{(T)} Q_{Y|X}^{(\boldsymbol{f}, \boldsymbol{g})} \right). \quad (17)$$

Then, the MM-$\chi^2$ algorithm is given in Algorithm 1. In practice, the loss $L_1(\alpha, \boldsymbol{f}, \boldsymbol{g})$, the related quantities $D$, and $\tilde{D}_1$ in Theorem 4.1 can be estimated by the empirical means of the features of samples. Detailed implementations are provided in the supplementary material. With the $\boldsymbol{f}^*$ and $\boldsymbol{g}^*$ computed by Algorithm 1, the predicted label $\hat{y}(x)$ for sample $x$ is given by the maximum a posterior (MAP) decision rule with $\hat{y}(x) = \arg\max_{y \in \mathcal{Y}} Q_{Y|X}^{(\boldsymbol{f}^*, \boldsymbol{g}^*)}(y|x)$.

## 4.2 REVISED FORMULATION 2 AND MM-HEL ALGORITHM

When Hellinger distance is chosen as the distance measure, we consider the discriminative model for the target distribution in the following form. For each $i = T, S$,

$$\sqrt{\tilde{Q}_{Y|X}^{(i, \boldsymbol{f}, \boldsymbol{h})}(y|x)} \triangleq \sqrt{P_Y^{(i)}(y)} \left( 1 + \boldsymbol{f}^{\mathrm{T}}(x)\boldsymbol{h}(y) \right). \quad (18)$$

This model is a deformation of (13), which makes the model trainable under Hellinger distance. Similarly, under the pre-trained feature extractor $\boldsymbol{f}^*$, the learned weights $\hat{\boldsymbol{h}}_i$ of each task $i = T, S$ can be defined as

$$\hat{\boldsymbol{h}}_i \triangleq \arg\min_{\boldsymbol{h}} H^2 \left( \hat{P}_{XY}^{(i)}, P_X^{(i)} \tilde{Q}_{Y|X}^{(i, \boldsymbol{f}^*, \boldsymbol{h})} \right). \quad (19)$$

Now $\hat{\boldsymbol{h}}_T$ and $\hat{\boldsymbol{h}}_S$ are the observations to generate the minimax solution, where the expectations can be defined as $\boldsymbol{h}_i(y) = \mathbb{E}[\hat{\boldsymbol{h}}_i(y)]$, $i = T, S$. The minimax problem can be defined as follows [cf. (8)]:

$$\boldsymbol{h}^* \triangleq \arg\min_{\boldsymbol{h}(\hat{\boldsymbol{h}}_T, \hat{\boldsymbol{h}}_S)} \max_{\boldsymbol{h}_S \in \mathcal{H}} \mathbb{E} \left[ H^2 \left( P_X^{(T)} \tilde{Q}_{Y|X}^{(T, \boldsymbol{f}^*, \boldsymbol{h}_T)}, P_X^{(T)} \tilde{Q}_{Y|X}^{(T, \boldsymbol{f}^*, \boldsymbol{h})} \right) \right], \quad (20)$$

where $\mathcal{H} \triangleq \left\{ \boldsymbol{h} : \frac{1}{2} \sum_{y \in \mathcal{Y}} \left\| \sqrt{P_Y^{(T)}(y)} \boldsymbol{\Lambda}_T^{\frac{1}{2}} \boldsymbol{h}_T(y) - \sqrt{P_Y^{(S)}(y)} \boldsymbol{\Lambda}_S^{\frac{1}{2}} \boldsymbol{h}(y) \right\|^2 \leq D^2 \right\}$, and $\boldsymbol{\Lambda}_i \triangleq \mathbb{E}_{P_X^{(i)}}[\boldsymbol{f}^*(X)\boldsymbol{f}^{*\mathrm{T}}(X)]$, for $i = T, S$.

We can directly apply Theorem 3.3 and obtain the following theorem.

**Theorem 4.2.** *When the empirical distrbutions* $\hat{P}_{XY}^{(T)}$ *and* $\hat{P}_{XY}^{(S)}$ *follow the density function (9), the minimax solution as defined in (20) is*

$$\boldsymbol{h}^*(y) = (1 - \tilde{\alpha}_2)\hat{\boldsymbol{h}}_T(y) + \tilde{\alpha}_2 \sqrt{P_Y^{(S)}(y)/P_Y^{(T)}(y)} \boldsymbol{\Lambda}_T^{-\frac{1}{2}} \boldsymbol{\Lambda}_S^{\frac{1}{2}} \hat{\boldsymbol{h}}_S(y),$$

*for all $y \in \mathcal{Y}$, where*

$$\tilde{\alpha}_2 = \frac{n_S}{n_T + n_S} \left( 1 - \frac{D}{\tilde{D}_2} J_{d|\mathcal{Y}|} \left( \frac{4 n_T n_S D \tilde{D}_2}{n_T + n_S} \right) \right), \quad (21)$$

*and* $\tilde{D}_2^2 \triangleq \frac{1}{2} \sum_{y \in \mathcal{Y}} \left\| \sqrt{P_Y^{(T)}(y)} \boldsymbol{\Lambda}_T^{\frac{1}{2}} \hat{\boldsymbol{h}}_T(y) - \sqrt{P_Y^{(S)}(y)} \boldsymbol{\Lambda}_S^{\frac{1}{2}} \hat{\boldsymbol{h}}_S(y) \right\|^2$.

---

**Algorithm 2** Minimax Hellinger-Algorithm (MM-Hel)

---

1: **Input:** target and source data samples $\{(x_l^{(i)}, y_l^{(i)})\}_{l=1}^{n_i}$ $(i = T, S)$
2: $(\boldsymbol{f}^*, \boldsymbol{h}_1^*, \boldsymbol{h}_2^*) \leftarrow \arg\min_{\boldsymbol{f}, \boldsymbol{h}_1, \boldsymbol{h}_2} L_2(\boldsymbol{f}, \boldsymbol{h}_1, \boldsymbol{h}_2)$
3: $\alpha \leftarrow \frac{n_S}{n_T + n_S} \left(1 - \frac{D}{D_2} J_{d|\mathcal{Y}|}\left(\frac{4 n_T n_S D \tilde{D}_2}{n_T + n_S}\right)\right)$
4: $\boldsymbol{h}^*(y) \leftarrow (1 - \alpha)\boldsymbol{h}_1^*(y) + \alpha \sqrt{\frac{P_Y^{(S)}(y)}{P_Y^{(T)}(y)}} \boldsymbol{\Lambda}_T^{-\frac{1}{2}} \boldsymbol{\Lambda}_S^{\frac{1}{2}} \boldsymbol{h}_2^*(y)$
5: **return** $\boldsymbol{f}^*, \boldsymbol{h}^*$

---

Similarly, we can design an algorithm based on Theorem 4.2. Note that we cannot apply the linearly combined training loss in the Hellinger distance setting due to the design of two different distribution models $\tilde{Q}_{Y|X}^{(T, \boldsymbol{f}, \boldsymbol{h})}$ and $\tilde{Q}_{Y|X}^{(S, \boldsymbol{f}, \boldsymbol{h})}$. We choose jointly training the shared feature extractor $\boldsymbol{f}$ and individual topmost layers of the target and source task. The training loss is defined as

$$L_2(\boldsymbol{f}, \boldsymbol{h}_1, \boldsymbol{h}_2) \triangleq H^2\left(\hat{P}_{XY}^{(T)}, P_X^{(T)} \tilde{Q}_{Y|X}^{(T, \boldsymbol{f}, \boldsymbol{h}_1)}\right) + H^2\left(\hat{P}_{XY}^{(S)}, P_X^{(S)} \tilde{Q}_{Y|X}^{(S, \boldsymbol{f}, \boldsymbol{h}_2)}\right). \tag{22}$$

Then, the MM-Hel algorithm is given in Algorithm 2. Similarly, we also provide the estimation of the related quantities in the supplementary material. With the computed $\boldsymbol{f}^*$ and $\boldsymbol{h}^*$, the predicted label $\hat{y}(x)$ for sample $x$ is given by the MAP decision rule with $\hat{y}(x) = \arg\max_{y \in \mathcal{Y}} \sqrt{\tilde{Q}_{Y|X}^{(T, \boldsymbol{f}^*, \boldsymbol{g}^*)}(y|x)}$.

## 5 EXPERIMENTS

To validate the theoretical analyses in Theorem 3.1 and Theorem 3.3, and the robustness of our algorithms, we conduct a series of experiments on common datasets for image recognition, including CIFAR-10 (Krizhevsky et al., 2009), Office-31 and Office-Caltech (Gong et al., 2012b) datasets. For convenience, different transfer settings are denoted by "source→target".

### 5.1 CIFAR-10

We conduct transfer learning experiments on CIFAR-10 dataset in order to verify the theoretical interpretations of the related factors in Remark 3.2, which mainly cover the sample size and task distance. Specifically, CIFAR-10 dataset contains 50 000 training images and 10 000 testing images in 10 classes. We first construct the source tasks and target task by dividing the original CIFAR-10 dataset into five disjoint sub-datasets, each containing two classes of the original data, which corresponds to a binary classification task. Then, we choose one as our target task (task 1), and use the other four as source tasks referred to as task 2, 3, 4, 5, where four corresponding transfer learning tasks are established.

In each transfer learning task, we use 2000 source images, with 1000 images per binary class. Target sample size $n$ is set as $n = 12, 20, 60, 200$ for four sub-tasks. Throughout this experiment, the feature $\boldsymbol{f}$ is of dimensionality $d = 10$, generated by GoogLeNet (Szegedy et al., 2015) pre-trained by ImageNet (Russakovsky et al., 2015), and followed by a fully connected layer.

The accuracies on the target testing images of MM-$\chi^2$ and MM-Hel algorithms are summarized in Table 1 and Table 2. In each task, target samples are randomly picked from the target training set. All the accuracies and standard deviations are reported over five random selections of target samples. In detail, we analyze the effect of target sample sizes as in figure 1. figure 1 shows the changes of the accuracies and coefficient $\tilde{\alpha}_1$ as defined in (16) (averaged over 5 tests) of different target sample sizes. The coefficient $\tilde{\alpha}_1$ represents how much the final model relies on the source task. It corresponds to our interpretation in Remark 3.2 that a larger target sample size could lead to less dependency on the source task, and meanwhile the accuracies become higher and more stable. We also analyze the effect of task distances as in figure 2. figure 2 shows the changes of the accuracies and task distance $\tilde{D}_2$ as defined in (21) (averaged over 5 tests) of different source tasks, where a larger task distance can lead to a worse accuracy and stability.

Table 1: Accuracies (%) of CIFAR-10 transfer learning tasks based on MM-$\chi^2$ algorithm, where $n_T$ represents the target sample size. The baseline is trained with merely target samples.

| Tasks | $n_T = 12$ | $n_T = 20$ | $n_T = 60$ | $n_T = 200$ |
|---|---|---|---|---|
| Baseline | 77.8±2.7 | 83.4±1.70 | 90.0±1.0 | 93.4±0.7 |
| $2 \to 1$ | 82.6±1.4 | 87.9±1.2 | 90.9±0.7 | 94.2±0.6 |
| $3 \to 1$ | 80.2±1.7 | 85.9±1.4 | 90.6±1.2 | 93.9±0.7 |
| $4 \to 1$ | 79.1±2.2 | 85.1±1.7 | 90.0±1.3 | 93.4±0.8 |
| $5 \to 1$ | 85.6±1.5 | 89.7±1.3 | 91.1±0.7 | 94.3±0.5 |

Table 2: Accuracies (%) of CIFAR-10 transfer learning tasks based on MM-Hel algorithm, where $n_T$ represents the target sample size and it shares the same baseline with Table 1.

| Tasks | $n_T = 12$ | $n_T = 20$ | $n_T = 60$ | $n_T = 200$ |
|---|---|---|---|---|
| $2 \to 1$ | 81.5±1.2 | 88.5±1.1 | 91.2±0.8 | 94.9±0.5 |
| $3 \to 1$ | 81.3±1.7 | 87.5±1.2 | 91.2±0.8 | 94.2±0.6 |
| $4 \to 1$ | 77.5±1.8 | 83.9±1.7 | 90.2±1.0 | 93.5±0.6 |
| $5 \to 1$ | 84.2±1.2 | 90.2±1.1 | 92.2±0.7 | 94.7±0.5 |

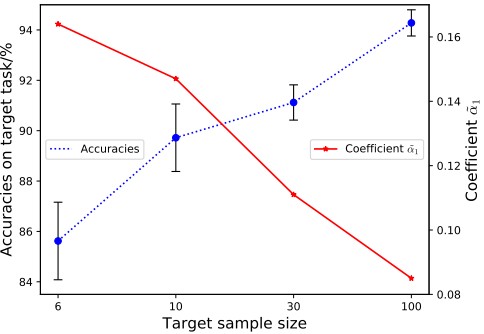

Figure 1: The accuracies and coefficient $\tilde{\alpha}_1$ in transferring task $5 \to 1$ based on MM-$\chi^2$ algorithm under different target sample sizes.

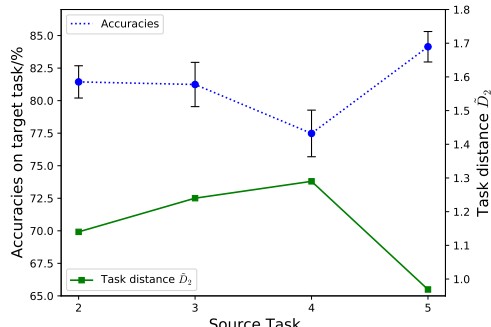

Figure 2: The accuracies and distance measure $\tilde{D}_2$ of target sample size $n_T = 12$ based on MM-Hel algorithm under different source tasks.

## 5.2 OFFICE-31

Office-31 dataset contains images of 31 categories with 3 sub-datasets, including Amazon (A, 2817 images), Dslr (D, 498 images), and Webcam (W, 795 images). Six transferring tasks can be established as A→D, A→W, D→W, D→A, W→A, and W→D. We adopt the transfer learning setting in (Tzeng et al., 2015), illustrated as follows. Specifically, 3 target samples per category are used for training, and the training sample size (per category) for source task is set to 20 or 8, depending on whether the source task is Amazon or not.

In this experiment, the feature $f$ is extracted by the VGG-16 (Simonyan & Zisserman, 2014) network pre-trained on the ImageNet, succeeded by fully connected layers, and the output is 64-dimensional. We introduce the UDDA (Motiian et al., 2017) algorithm as the typical baseline and the iterative linear combination method (Tong et al., 2021) (ILCM) for comparison, which employs a similar linear combination method.

Table 3 summarizes test accuracies under different transfer settings, where all reported accuracies and standard deviations are averaged over five train-test splits. The results indicate that our algorithms generally have higher robustness and competitive accuracies.

## 5.3 OFFICE-CALTECH

Office-Caltech dataset is composed of 10 categories, divided as four sub-datasets: Amazon (A, 958 images), Caltech (C, 1123 images), Webcam (W, 295 images), and Dslr (D, 157 images). We focus on the six transfer settings depending on C, i.e., A→C, W→C, D→C, C→A, C→W, and C→D. The train-test split is as introduced in (Gong et al., 2012a). The feature $f$ is based on the pre-trained DeCAF network (Donahue et al., 2014), succeeded by fully connected layers, and the output dimension is $d = 10$.

Table 4 shows the performances in comparison with CPNN (Ding et al., 2018) and ILCM (Tong et al., 2021) algorithms, where CPNN is chosen as the baseline to be consistent with ILCM.

Table 3: Test accuracies for target tasks under different transfer settings on *Office-31*.

| Method | A→D | A→W | D→W | D→A | W→A | W→D |
|---|---|---|---|---|---|---|
| UDDA | 89.0±1.2 | **88.2**±1.0 | 96.4±0.8 | 71.8±0.5 | 72.1±1.0 | **97.6**±0.4 |
| ILCM | 90.0±1.4 | 87.3±1.1 | 96.5±1.0 | 72.4±1.0 | 72.1±0.9 | 97.2±0.4 |
| MM-$\chi^2$ (Ours) | **90.2**±0.6 | 87.5±0.5 | **96.7**±0.5 | **72.9**±0.8 | **72.2**±0.6 | **97.6**±**0.3** |
| MM-Hel (Ours) | 89.8±**0.5** | 88.1±**0.4** | 96.0±**0.5** | 72.2±**0.5** | 72.0±**0.4** | 97.5±0.4 |

Table 4: Test accuracies for target tasks under different transfer settings on *Office-Caltech*.

| Method | A→C | W→C | D→C | C→A | C→W | C→D |
|---|---|---|---|---|---|---|
| CPNN | 74.3±0.6 | 72.1±0.8 | 66.6±0.8 | 86.2±0.5 | 86.0±0.5 | 79.9±0.7 |
| ILCM | **80.3**±0.7 | 72.9±0.7 | **72.2**±0.9 | 88.4±0.7 | 85.9±0.5 | 83.5±0.9 |
| MM-$\chi^2$ (Ours) | 79.9±0.7 | **73.5**±0.5 | 71.7±0.7 | **90.1**±0.5 | **86.8**±0.5 | **84.0**±0.7 |
| MM-Hel (Ours) | 79.2±**0.3** | 72.7±**0.5** | 72.1±**0.6** | 88.5±**0.4** | 85.5±**0.4** | 83.5±**0.5** |

# 6 RELATED WORKS

## 6.1 MINIMAX ESTIMATOR OF BOUNDED NORMAL MEAN

Minimax estimator is a significant theme in statistical decision theory, which deals with the problem of estimating a deterministic parameter in a certain family (Hodges & Lehmann, 2012). Under the special setting of bounded normal mean, many works study the analytical solution when the centers of Gaussian observations are restricted, including analytical solution for 1-dimensional observations (Casella & Strawderman, 1981), high-dimensional observations (Berry, 1990; Marchand & Perron, 2002). Moreover, the objective function can also be measured by norms other than mean square error (Bischoff et al., 1995), which allows the applications in machine learning scenarios and help derive the solutions of this paper's formulations.

## 6.2 SELECTION OF DISTANCE MEASURE

In this paper, we select $\chi^2$-distance and Hellinger distance as the distance measure in (3), which help analytically solve the minimax problem. These two measurements are both from the family of $f$-divergence (Csiszár & Shields, 2004) and are widely-used in machine learning. Specifically, $\chi^2$-distance can lead to the typical alternating conditional expectation algorithm (Xu & Huang, 2020). Hellinger distance is also used to evaluate the domain adaptation in transfer learning (Baktashmotlagh et al., 2014; 2016). Moreover, most existing measurements with non-linear functions, e.g, K-L divergence containing the logarithm function, could be ill-defined, as explained in section 2.2.

## 6.3 MINIMAX TRANSFER LEARNING AND ROBUSTNESS

Minimax principle has been widely-used in transfer learning to promote the robustness of algorithms (Verdu & Poor, 1984). The most common empirical method is connected to the adversarial learning methods (Shafahi et al., 2019), including maximizing the training loss of adversarial classifier (Tzeng et al., 2017) and maximizing the discrepancy between classifiers' outputs (Saito et al., 2018). Meanwhile, transfer learning settings can naturally imply minimax optimization problems in view of the relationship between target and source tasks (Zhang et al., 2019). Recent researches reveal that the maximization can succeed over the constraints on the distribution shift (Lei et al., 2021), the similarity between neaural network parameters (Kalan et al., 2020), and optional source tasks (Cai & Wei, 2021).

# 7 CONCLUSION

This paper introduces a minimax framework for transfer learning based on the assumption of task distance. We provide the analytical solution of the minimax problem and characterize the roles of sample sizes, task distance, and model dimensionality in knowledge transferring. In addition, we develop robust transfer learning algorithms based on theoretical analyses. Experiments on practical tasks show the robustness and effectiveness of our proposed algorithms.

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

## A  APPROXIMATION OF K-L DIVERGENCE

Firstly, we provide the following approximation of K-L divergence, under the assumption that $R(z)$, $P(z)$, and $Q(z)$ are close to each other.

**Lemma A.1.** *Suppose that $|P(z) - R(z)| < \epsilon$ and $|Q(z) - R(z)| < \epsilon$ for each $z \in \mathcal{Z}$, where $\epsilon/|\mathcal{Z}| \ll 1$, K-L divergence between $P(z)$ and $Q(z)$ can have the following approximation*

$$D\left(P\|Q\right) = \frac{1}{2}\chi_R^2(P, Q) + O(\epsilon^3). \tag{23}$$

*Proof.*

$$
\begin{aligned}
D\left(P\|Q\right) &= \sum_{z \in \mathcal{Z}} -P(z) \log \frac{Q(z)}{P(z)} \\
&= \sum_{z \in \mathcal{Z}} -P(z) \log \left(1 + \frac{Q(z) - P(z)}{P(z)}\right) \\
&= \sum_{z \in \mathcal{Z}} -P(z) \left(\frac{Q(z) - P(z)}{P(z)} - \frac{(Q(z) - P(z))^2}{2P^2(z)}\right) + O(\epsilon^3) \\
&= \sum_{z \in \mathcal{Z}} \frac{(Q(z) - P(z))^2}{2P(z)} + O(\epsilon^3) \\
&= \sum_{z \in \mathcal{Z}} \frac{(Q(z) - P(z))^2}{2(R(z) + P(z) - R(z))} + O(\epsilon^3) \\
&= \frac{1}{2} \sum_{z \in \mathcal{Z}} \frac{(Q(z) - P(z))^2}{R(z)} + O(\epsilon^3).
\end{aligned}
$$

$\square$

Secondly, we consider Hellinger distance is closely connected with K-L divergence as the lower bound, which is explained in the following lemma.

**Lemma A.2.** *Let $P(z)$, and $Q(z)$ be the distribution supported by $\mathcal{Z}$. We have*

$$D\left(P\|Q\right) \geq 2H^2(P, Q). \tag{24}$$

The proof of this lemma is easy to find and omitted here.

# B PROOF OF THEOREM 3.1 AND THEOREM 3.3

First, the regular definition of the modified Bessel functions of the first kind is

$$I_\nu(x) = \sum_{m=0}^{\infty} \frac{1}{m!\Gamma(m + \nu + 1)} \left(\frac{x}{2}\right)^{(2m+\nu)},$$

where $\Gamma(\cdot)$ denotes the gamma function.

To solve the minimax problem (6) and (8), we apply the minimax estimation of a bounded normal mean vector Berry (1990). The following lemma is a direct extension of the bounded normal mean results.

**Lemma B.1.** *Given two observations $\boldsymbol{y} \sim \mathcal{N}(\boldsymbol{x}, \sigma_1^2 \boldsymbol{I}_k)$ and $\boldsymbol{w} \sim \mathcal{N}(\boldsymbol{z}, \sigma_2^2 \boldsymbol{I}_k)$, where $\boldsymbol{I}_k$ denotes the $k \times k$ identity matrix, their centers satisfy $\|\boldsymbol{x} - \boldsymbol{z}\| \leq D$, and $D/\sqrt{\sigma_1^2 + \sigma_2^2} \leq \sqrt{k}$. Let $\hat{\boldsymbol{x}}^*$ be the minimax estimator for the following minimax problem, e.g.,*

$$\hat{\boldsymbol{x}}^* = \arg\min_{\hat{\boldsymbol{x}}(\boldsymbol{y},\boldsymbol{w})} \max_{\boldsymbol{z}:\|\boldsymbol{x}-\boldsymbol{z}\|^2 \leq D^2} \mathbb{E}[\|\hat{\boldsymbol{x}}(\boldsymbol{y},\boldsymbol{w}) - \boldsymbol{x}\|^2]. \tag{25}$$

*Then, the expression of $\hat{\boldsymbol{x}}^*$ is*

$$\hat{\boldsymbol{x}}^* = \frac{\sigma_1^2}{\sigma_1^2 + \sigma_2^2}\boldsymbol{w} + \frac{\sigma_2^2}{\sigma_1^2 + \sigma_2^2}\boldsymbol{y} + \frac{\sigma_1^2}{\sigma_1^2 + \sigma_2^2} \frac{I_{\frac{k}{2}}\left(\frac{D}{\sigma_1^2+\sigma_2^2}\|\boldsymbol{y} - \boldsymbol{w}\|\right)}{I_{\frac{k}{2}-1}\left(\frac{D}{\sigma_1^2+\sigma_2^2}\|\boldsymbol{y} - \boldsymbol{w}\|\right)} \frac{D}{\|\boldsymbol{y} - \boldsymbol{w}\|}(\boldsymbol{y} - \boldsymbol{w}). \tag{26}$$

*Proof.* First, we derive the posterior MMSE estimator for $\boldsymbol{x}$ under the uniform prior distribution on the surface of the sphere.

**Lemma B.2** (MMSE estimator). *When the means and variances are finite, the MMSE estimator for parameter $x$ of observation $y$ is uniquely defined and is given by*

$$\hat{x}(y) = \mathbb{E}[x|y]. \tag{27}$$

The likelihood of the observation $\boldsymbol{y}$, $\boldsymbol{w}$ is

$$\mathbb{P}(\boldsymbol{y}, \boldsymbol{w}|\boldsymbol{x}, \boldsymbol{z}) = \mathbb{P}(\boldsymbol{y}|\boldsymbol{x})\,\mathbb{P}(\boldsymbol{w}|\boldsymbol{z}) \propto \exp\left(-\frac{(\boldsymbol{y} - \boldsymbol{x})^2}{2\sigma_1^2}\right)\exp\left(-\frac{(\boldsymbol{z} - \boldsymbol{w})^2}{2\sigma_2^2}\right). \tag{28}$$

Let $\pi(\boldsymbol{x}|\boldsymbol{z})$ be the uniform prior distribution on the surface of the sphere in $k$ dimensions with center at $\boldsymbol{z}$ and radius $D$. Let $\boldsymbol{t} \triangleq \boldsymbol{x} - \boldsymbol{z}$, and we denote the prior as $\pi(\boldsymbol{t}) = \mathbb{1}_{\{\|\boldsymbol{t}\|=D\}}(\boldsymbol{t})$.

Under such a prior distribution,

$$\mathbb{P}(\boldsymbol{y}, \boldsymbol{w}|\boldsymbol{x}) \propto \int_{\mathbb{R}^k} \exp\left(-\frac{(\boldsymbol{y} - \boldsymbol{x})^2}{2\sigma_1^2}\right)\exp\left(-\frac{(\boldsymbol{z} - \boldsymbol{w})^2}{2\sigma_2^2}\right) \mathbb{1}_{\{\|\boldsymbol{x}-\boldsymbol{z}\|=D\}}(\boldsymbol{x} - \boldsymbol{z})\mathrm{d}\boldsymbol{z}. \tag{29}$$

Then, the posterior distribution is

$$\begin{aligned}
\mathbb{P}(\boldsymbol{x}|\boldsymbol{y}, \boldsymbol{w}) &\propto \int_{\mathbb{R}^k} \exp\left(-\frac{(\boldsymbol{y} - \boldsymbol{x})^2}{2\sigma_1^2}\right)\exp\left(-\frac{(\boldsymbol{z} - \boldsymbol{w})^2}{2\sigma_2^2}\right)\mathbb{1}_{\{\|\boldsymbol{x}-\boldsymbol{z}\|=D\}}(\boldsymbol{z})\mathrm{d}\boldsymbol{z} \\
&\propto \int_{\mathbb{R}^k} \exp\left(-\frac{(\boldsymbol{x} - \boldsymbol{y})^2}{2\sigma_1^2}\right)\exp\left(-\frac{(\boldsymbol{x} - (\boldsymbol{w} + \boldsymbol{t}))^2}{2\sigma_2^2}\right)\mathbb{1}_{\{\|\boldsymbol{t}\|=D\}}(\boldsymbol{t})\mathrm{d}\boldsymbol{t} \\
&\propto \int_{\mathbb{R}^k} \exp\left(-\frac{1}{2}\left(\frac{1}{\sigma_1^2} + \frac{1}{\sigma_2^2}\right)\left(\boldsymbol{x} - \frac{\sigma_2^2\boldsymbol{y} + \sigma_1^2(\boldsymbol{w} + \boldsymbol{t})}{\sigma_1^2 + \sigma_2^2}\right)^2\right) \\
&\qquad \cdot \exp\left(-\frac{1}{2}\frac{(\boldsymbol{y} - (\boldsymbol{w} + \boldsymbol{t}))^2}{\sigma_1^2 + \sigma_2^2}\right)\mathbb{1}_{\{\|\boldsymbol{t}\|=D\}}(\boldsymbol{t})\mathrm{d}\boldsymbol{t}.
\end{aligned} \tag{30}$$

Therefore, the Bayes estimator of the posterior is

$$
\hat{x}^* = \mathbb{E}\left[x|y, w\right]
$$

$$
= A \int_{\mathbb{R}^k} x \int_{\mathbb{R}^k} \exp\left(-\frac{1}{2}\left(\frac{1}{\sigma_1^2} + \frac{1}{\sigma_2^2}\right)\left(x - \frac{\sigma_2^2 y + \sigma_1^2(w + t)}{\sigma_1^2 + \sigma_2^2}\right)^2\right)
$$

$$
\cdot \exp\left(-\frac{1}{2}\frac{(y - (w + t))^2}{\sigma_1^2 + \sigma_2^2}\right) \mathbb{1}_{\{\|t\|=D\}}(t)\mathrm{d}t\mathrm{d}x
$$

$$
= A \int_{\mathbb{R}^k} \int_{\mathbb{R}^k} x \exp\left(-\frac{1}{2}\left(\frac{1}{\sigma_1^2} + \frac{1}{\sigma_2^2}\right)\left(x - \frac{\sigma_2^2 y + \sigma_1^2(w + t)}{\sigma_1^2 + \sigma_2^2}\right)^2\right)
$$

$$
\cdot \exp\left(-\frac{1}{2}\frac{(y - (w + t))^2}{\sigma_1^2 + \sigma_2^2}\right) \mathbb{1}_{\{\|t\|=D\}}(t)\mathrm{d}x\mathrm{d}t
$$

$$
= A' \int_{\mathbb{R}^k} \frac{\sigma_2^2 y + \sigma_1^2(w + t)}{\sigma_1^2 + \sigma_2^2} \exp\left(-\frac{1}{2}\frac{(y - (w + t))^2}{\sigma_1^2 + \sigma_2^2}\right) \mathbb{1}_{\{\|t\|=D\}}(t)\mathrm{d}t
$$

$$
= \frac{\sigma_2^2 y + \sigma_1^2 w}{\sigma_1^2 + \sigma_2^2} + \frac{\sigma_1^2}{\sigma_1^2 + \sigma_2^2} A' \int_{\mathbb{R}^k} t \exp\left(-\frac{1}{2}\frac{(y - (w + t))^2}{\sigma_1^2 + \sigma_2^2}\right) \mathbb{1}_{\{\|t\|=D\}}(t)\mathrm{d}t
$$

$$
= \frac{\sigma_2^2 y + \sigma_1^2 w}{\sigma_1^2 + \sigma_2^2} + \frac{\sigma_1^2}{\sigma_1^2 + \sigma_2^2} \frac{I_{\frac{k}{2}}(\frac{D}{\sigma_1^2+\sigma_2^2}\|y - w\|)}{I_{\frac{k}{2}-1}(\frac{D}{\sigma_1^2+\sigma_2^2}\|y - w\|)} \frac{D}{\|y - w\|}(y - w), \tag{31}
$$

where $A$ and $A'$ are the normalization constants.

Then, we will prove that $\hat{x}^*$ is the minimax estimator. Specifically , we prove that $\sigma_2^2 y + \sigma_1^2 w$ is independent of $y - w$. Since both two r.v.s are normal, we only need to prove $\mathrm{Cov}(\sigma_2^2 y + \sigma_1^2 w, y - w) = 0$, i.e.,

$$
\mathrm{Cov}(\sigma_2^2 y + \sigma_1^2 w, y - w) = \sigma_2^2 \mathrm{Var}(y) + \sigma_1^2 \mathrm{Var}(w) = \sigma_1^2 \sigma_2^2 - \sigma_1^2 \sigma_2^2 = 0. \tag{32}
$$

We then define the risk function

$$
R_{\hat{x}^*}(t) \triangleq \mathbb{E}\left[\left(\frac{\sigma_2^2 y + \sigma_1^2 w}{\sigma_1^2 + \sigma_2^2} + \frac{\sigma_1^2}{\sigma_1^2 + \sigma_2^2} \frac{I_{\frac{k}{2}}(\frac{D}{\sigma_1^2+\sigma_2^2}\|y - w\|)}{I_{\frac{k}{2}-1}(\frac{D}{\sigma_1^2+\sigma_2^2}\|y - w\|)} \frac{D}{\|y - w\|}(y - w) - x\right)^2\right]
$$

$$
= \frac{2\sigma_1^2 \sigma_2^2}{\sigma_1^2 + \sigma_2^2} + \left(\frac{\sigma_1^2}{\sigma_1^2 + \sigma_2^2}\right)^2
$$

$$
\cdot \mathbb{E}_{y-w\sim\mathcal{N}(t,\sigma_1^2+\sigma_2^2)}\left[\left(\frac{I_{\frac{k}{2}}(\frac{D}{\sigma_1^2+\sigma_2^2}\|y - w\|)}{I_{\frac{k}{2}-1}(\frac{D}{\sigma_1^2+\sigma_2^2}\|y - w\|)} \frac{D}{\|y - w\|}(y - w) - t\right)^2\right]. \tag{33}
$$

**Lemma B.3** (Minimax Theorem Marchand & Perron (2002)). *The unique Bayes estimator $\hat{x}^*$ is also the unique minimax estimator when*

$$
\max_{t} R_{\hat{x}^*}(t) = \int R_{\hat{x}^*}(t)\mathrm{d}\pi(t).
$$

Let $R'(t) \triangleq \mathbb{E}_{(y-w)\sim\mathcal{N}(t,\sigma_1^2+\sigma_2^2)}\left[\left(\frac{I_{\frac{k}{2}}(\frac{D}{\sigma_1^2+\sigma_2^2}\|y-w\|)}{I_{\frac{k}{2}-1}(\frac{D}{\sigma_1^2+\sigma_2^2}\|y-w\|)} \frac{D}{\|y-w\|}(y - w) - t\right)^2\right].$

**Lemma B.4** (Berry (1990)). *When $D/\sqrt{\sigma_1^2 + \sigma_2^2} \le \sqrt{k}$,*

$$
\max_{t} R'(t) = \int R'(t)\mathrm{d}\pi(t) \tag{34}
$$

Based on Lemma B.3 and Lemma B.4, $\hat{x}^*$ is the minimax estimator. □

Lemma B.1 reveals that when two Gaussian observations contain a prior knowledge that their centers have a maximum distance, the optimal estimator is a linear combination of the observations, where the combining coefficient is related to the variances of the two observations, the maximum center distance, and the dimension of observations.

For Formulation 1 (6), we can define a random vector $\boldsymbol{u}, \boldsymbol{v} \in \mathbb{R}^{|\mathcal{X}||\mathcal{Y}|}$, where for all $(x, y) \in \mathcal{X} \times \mathcal{Y}$,

$$\boldsymbol{u}(x, y) \triangleq \frac{\hat{P}_{XY}^{(T)}(x, y)}{\sqrt{R(x, y)}}, \tag{35}$$

and

$$\boldsymbol{v}(x, y) \triangleq \frac{\hat{P}_{XY}^{(S)}(x, y)}{\sqrt{R(x, y)}}. \tag{36}$$

Their centers are $\boldsymbol{u}_0(x, y) \triangleq \frac{P_{XY}^{(T)}(x,y)}{\sqrt{R(x,y)}}$ and $\boldsymbol{v}_0(x, y) \triangleq \frac{P_{XY}^{(S)}(x,y)}{\sqrt{R(x,y)}}$. According to (7), we have $\boldsymbol{u} \sim \mathcal{N}(\boldsymbol{u}_0, \frac{1}{n_T} \boldsymbol{I}_{|\mathcal{X}||\mathcal{Y}|})$ and $\boldsymbol{v} \sim \mathcal{N}(\boldsymbol{v}_0, \frac{1}{n_S} \boldsymbol{I}_{|\mathcal{X}||\mathcal{Y}|})$ , and problem (6) can be re-defined as

$$\min_{\boldsymbol{w}(\boldsymbol{u}, \boldsymbol{v})} \max_{\boldsymbol{v}_0 : \|\boldsymbol{v}_0 - \boldsymbol{u}_0\|^2 \leq D^2} \mathbb{E}\left[\|\boldsymbol{u}_0 - \boldsymbol{w}\|^2\right]. \tag{37}$$

With Lemma B.1, we derive Theorem 3.1.

For Formulation 2 (8), we can define a random vector $\boldsymbol{u}, \boldsymbol{v} \in \mathbb{R}^{|\mathcal{X}||\mathcal{Y}|}$, where for all $(x, y) \in \mathcal{X} \times \mathcal{Y}$,

$$\boldsymbol{u}(x, y) \triangleq \sqrt{\hat{P}_{XY}^{(T)}(x, y)}, \tag{38}$$

and

$$\boldsymbol{v}(x, y) \triangleq \sqrt{\hat{P}_{XY}^{(S)}(x, y)}. \tag{39}$$

Their centers are $\boldsymbol{u}_0(x, y) \triangleq \sqrt{P_{XY}^{(T)}(x, y)}$ and $\boldsymbol{v}_0(x, y) \triangleq \sqrt{P_{XY}^{(S)}(x, y)}$. According to (9), we have $\boldsymbol{u} \sim \mathcal{N}(\boldsymbol{u}_0, \frac{1}{2n_T} \boldsymbol{I}_{|\mathcal{X}||\mathcal{Y}|})$ and $\boldsymbol{v} \sim \mathcal{N}(\boldsymbol{v}_0, \frac{1}{2n_S} \boldsymbol{I}_{|\mathcal{X}||\mathcal{Y}|})$ , and problem (8) can be re-defined as

$$\min_{\boldsymbol{w}(\boldsymbol{u}, \boldsymbol{v})} \max_{\boldsymbol{v}_0 : \frac{1}{2}\|\boldsymbol{v}_0 - \boldsymbol{u}_0\|^2 \leq D^2} \mathbb{E}\left[\frac{1}{2}\|\boldsymbol{u}_0 - \boldsymbol{w}\|^2\right]. \tag{40}$$

With Lemma B.1, we derive Theorem 3.3.

## C  INTERPRETATION OF THEOREM 3.1

Firstly, Remark 3.2 leads to that $\alpha_1 \simeq \frac{n_S}{n_T + n_S}\left(1 - \frac{n_T n_S D^2}{(n_S + n_T)|\mathcal{X}||\mathcal{Y}|}\right)$. Considering that $|\mathcal{X}||\mathcal{Y}| \ll D^2$, we have

$$\alpha_1 \simeq \frac{n_S}{n_T + n_S} \frac{1}{1 + \frac{n_T n_S D^2}{(n_S + n_T)|\mathcal{X}||\mathcal{Y}|}} = \frac{\frac{|\mathcal{X}||\mathcal{Y}|}{n_T}}{\frac{|\mathcal{X}||\mathcal{Y}|}{n_T} + \frac{|\mathcal{X}||\mathcal{Y}|}{n_S} + D^2}, \tag{41}$$

which is close to Eq.(6) in Tong et al. (2021).

A geometric explanation for Theorem 3.1 can be depicted in figure 3, where the entire space represents all the distributions supported by $\mathcal{X} \times \mathcal{Y}$. Two balls centered at the target and source distributions represent the empirical distribution sets $\mathcal{P}_{n_T}$ and $\mathcal{P}_{n_S}$. In particular, the radii of the balls show the variances of the empirical distributions, which are inversely proportional to the sample sizes. The area with a deeper color contains those paracentral empirical distributions, which have higher probability according to (7). In addition, the distance between the centers is determined by the task distance assumption. The minimax problem (6) is meant to find estimator $Q_{XY}^{(1)}$, which is closest to the target distribution in average, where the dash line implies the linear family of $\hat{P}_{XY}^{(T)}$ and $\hat{P}_{XY}^{(S)}$. When the target sample size increases, the blue ball will shrink. When distance $D$ increases, two balls will be farther away. Then, $\hat{P}_{XY}^{(T)}$ would be closer to the target distribution, which implies a smaller coefficient $\alpha_1$.

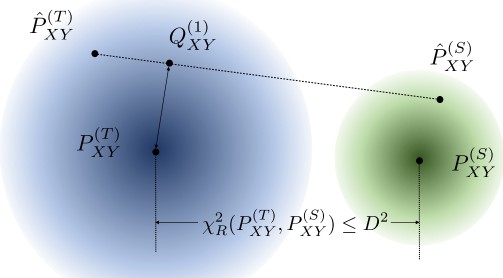

Figure 3: A geometrical explanation of the minimax setting (6). Two balls centered at the underlying distributions represent all possible empirical distributions.

## D    PROOF OF THEOREM 4.1 AND THEOREM 4.2

Without loss of generality, we assume $\mathbb{E}_{P_X^{(T)}}[\boldsymbol{f}(X)] = \boldsymbol{0}$. It can be easily guaranteed by deducting the average feature when computing the feature of each $x$.

We denote $\boldsymbol{\Lambda}_T \triangleq \mathbb{E}_{P_X^{(T)}}[\boldsymbol{f}^*(X)\boldsymbol{f}^{*\mathrm{T}}(X)]$ and $\boldsymbol{\Lambda}_S \triangleq \mathbb{E}_{P_X^{(S)}}[\boldsymbol{f}^*(X)\boldsymbol{f}^{*\mathrm{T}}(X)]$.

### D.1    REVISED FORMULATION 1

For revised Formulation 1 (15), first, we give the expression of $\hat{\boldsymbol{g}}_i$ as defined in (14) Tong et al. (2021)

$$\hat{\boldsymbol{g}}_i(y) = \frac{1}{P_Y^{(T)}(y)}\boldsymbol{\Lambda}_T^{-1}\left(\sum_{x\in\mathcal{X}}\hat{P}_{XY}^{(i)}(x,y)\boldsymbol{f}^*(x)\right). \tag{42}$$

Then, we can define a random vector $\boldsymbol{u}, \boldsymbol{v} \in \mathbb{R}^{d|\mathcal{Y}|}$, where

$$\boldsymbol{u} \triangleq \left[\sqrt{P_Y^{(T)}(1)}\boldsymbol{\Lambda}_T^{\frac{1}{2}}\hat{\boldsymbol{g}}_T^{\mathrm{T}}(1), \cdots, \sqrt{P_Y^{(T)}(|\mathcal{Y}|)}\boldsymbol{\Lambda}_T^{\frac{1}{2}}\hat{\boldsymbol{g}}_T^{\mathrm{T}}(|\mathcal{Y}|)\right]^{\mathrm{T}}, \tag{43}$$

and

$$\boldsymbol{v} \triangleq \left[\sqrt{P_Y^{(T)}(1)}\boldsymbol{\Lambda}_T^{\frac{1}{2}}\hat{\boldsymbol{g}}_S^{\mathrm{T}}(1), \cdots, \sqrt{P_Y^{(T)}(|\mathcal{Y}|)}\boldsymbol{\Lambda}_T^{\frac{1}{2}}\hat{\boldsymbol{g}}_S^{\mathrm{T}}(|\mathcal{Y}|)\right]^{\mathrm{T}}. \tag{44}$$

Their centers are

$$\boldsymbol{u}_0 \triangleq \left[\sqrt{P_Y^{(T)}(1)}\boldsymbol{\Lambda}_T^{\frac{1}{2}}\boldsymbol{g}_T^{\mathrm{T}}(1), \cdots, \sqrt{P_Y^{(T)}(|\mathcal{Y}|)}\boldsymbol{\Lambda}_T^{\frac{1}{2}}\boldsymbol{g}_T^{\mathrm{T}}(|\mathcal{Y}|)\right]^{\mathrm{T}}, \tag{45}$$

and

$$\boldsymbol{v}_0 \triangleq \left[\sqrt{P_Y^{(T)}(1)}\boldsymbol{\Lambda}_T^{\frac{1}{2}}\boldsymbol{g}_S^{\mathrm{T}}(1), \cdots, \sqrt{P_Y^{(T)}(|\mathcal{Y}|)}\boldsymbol{\Lambda}_T^{\frac{1}{2}}\boldsymbol{g}_S^{\mathrm{T}}(|\mathcal{Y}|)\right]^{\mathrm{T}}. \tag{46}$$

With (7), we have

$$\boldsymbol{u} \sim \mathcal{N}(\boldsymbol{u}_0, \frac{1}{n_T}\boldsymbol{I}_{d|\mathcal{Y}|}), \ \boldsymbol{v} \sim \mathcal{N}(\boldsymbol{v}_0, \frac{1}{n_S}\boldsymbol{I}_{d|\mathcal{Y}|}). \tag{47}$$

Problem (15) can be re-defined as

$$\min_{\boldsymbol{w}(\boldsymbol{u},\boldsymbol{v})} \max_{\boldsymbol{v}_0 : \|\boldsymbol{v}_0 - \boldsymbol{u}_0\|^2 \le D^2} \mathbb{E}\left[\|\boldsymbol{u}_0 - \boldsymbol{w}\|^2\right], \tag{48}$$

and thus Theorem 4.1 is proved.

## D.2 REVISED FORMULATION 2

For revised Formulation 2 (15), we give the expression of $\hat{\boldsymbol{h}}_i$ as defined in (19),

$$\hat{\boldsymbol{h}}_i(y) = \frac{1}{\sqrt{P_Y^{(i)}(y)}} \boldsymbol{\Lambda}_i^{-1} \left( \sum_{x \in \mathcal{X}} \sqrt{\hat{P}_{XY}^{(i)}(x, y) P_X^{(i)}(x)} \boldsymbol{f}^*(x) \right). \tag{49}$$

Then, we can define a random vector $\boldsymbol{u}, \boldsymbol{v} \in \mathbb{R}^{d|\mathcal{Y}|}$, where

$$\boldsymbol{u} \triangleq \left[ \sqrt{P_Y^{(T)}(1)} \boldsymbol{\Lambda}_T^{\frac{1}{2}} \hat{\boldsymbol{h}}_T^{\mathrm{T}}(1), \cdots, \sqrt{P_Y^{(T)}(|\mathcal{Y}|)} \boldsymbol{\Lambda}_T^{\frac{1}{2}} \hat{\boldsymbol{h}}_T^{\mathrm{T}}(|\mathcal{Y}|) \right]^{\mathrm{T}}, \tag{50}$$

and

$$\boldsymbol{v} \triangleq \left[ \sqrt{P_Y^{(S)}(1)} \boldsymbol{\Lambda}_S^{\frac{1}{2}} \hat{\boldsymbol{h}}_S^{\mathrm{T}}(1), \cdots, \sqrt{P_Y^{(S)}(|\mathcal{Y}|)} \boldsymbol{\Lambda}_S^{\frac{1}{2}} \hat{\boldsymbol{h}}_S^{\mathrm{T}}(|\mathcal{Y}|) \right]^{\mathrm{T}}. \tag{51}$$

Their centers are

$$\boldsymbol{u}_0 \triangleq \left[ \sqrt{P_Y^{(T)}(1)} \boldsymbol{\Lambda}_T^{\frac{1}{2}} \boldsymbol{h}_T^{\mathrm{T}}(1), \cdots, \sqrt{P_Y^{(T)}(|\mathcal{Y}|)} \boldsymbol{\Lambda}_T^{\frac{1}{2}} \boldsymbol{h}_T^{\mathrm{T}}(|\mathcal{Y}|) \right]^{\mathrm{T}}, \tag{52}$$

and

$$\boldsymbol{v}_0 \triangleq \left[ \sqrt{P_Y^{(S)}(1)} \boldsymbol{\Lambda}_S^{\frac{1}{2}} \boldsymbol{h}_S^{\mathrm{T}}(1), \cdots, \sqrt{P_Y^{(S)}(|\mathcal{Y}|)} \boldsymbol{\Lambda}_S^{\frac{1}{2}} \boldsymbol{h}_S^{\mathrm{T}}(|\mathcal{Y}|) \right]^{\mathrm{T}}. \tag{53}$$

With (7), we have

$$\boldsymbol{u} \sim \mathcal{N}(\boldsymbol{u}_0, \frac{1}{2n_T} \boldsymbol{I}_{d|\mathcal{Y}|}), \ \boldsymbol{v} \sim \mathcal{N}(\boldsymbol{v}_0, \frac{1}{2n_S} \boldsymbol{I}_{d|\mathcal{Y}|}). \tag{54}$$

Problem (15) can be re-defined as

$$\min_{\boldsymbol{w}(\boldsymbol{u}, \boldsymbol{v})} \max_{\boldsymbol{v}_0 : \frac{1}{2} \|\boldsymbol{v}_0 - \boldsymbol{u}_0\|^2 \le D^2} \mathbb{E} \left[ \frac{1}{2} \|\boldsymbol{u}_0 - \boldsymbol{w}\|^2 \right], \tag{55}$$

and the optimal estimator is

$$(1 - \tilde{\alpha}_2) \boldsymbol{u} + \tilde{\alpha}_2. \tag{56}$$

Note that $\boldsymbol{w}$ here refers to the vector

$$\left[ \sqrt{P_Y^{(T)}(1)} \boldsymbol{\Lambda}_T^{\frac{1}{2}} \boldsymbol{h}^{\mathrm{T}}(1), \cdots, \sqrt{P_Y^{(T)}(|\mathcal{Y}|)} \boldsymbol{\Lambda}_T^{\frac{1}{2}} \boldsymbol{h}^{\mathrm{T}}(|\mathcal{Y}|) \right]^{\mathrm{T}}.$$

Thus, Theorem 4.1 is proved.

## E DETAILS OF ALGORITHM IMPLEMENTATIONS

### E.1 ALGORITHM 1

Here we provide the details of the loss function $L_1(\alpha, \boldsymbol{f}, \boldsymbol{g})$ in line 4 of Algorithm 1, the quantity $D$ and the quantity $\tilde{D}_1$ in line 5. Our main procedures follow the results in Xu & Huang (2020)

As for the zero-mean assumption of $\boldsymbol{f}^*$, which results in zero-mean weights $\boldsymbol{g}$, we first define $\tilde{\boldsymbol{f}}(X) \triangleq \boldsymbol{f}(X) - \mathbb{E}_{\hat{P}_X^{(T)}}[\boldsymbol{f}(X)]$ and $\tilde{\boldsymbol{g}}(Y) \triangleq \boldsymbol{g}(Y) - \mathbb{E}_{\hat{P}_Y^{(T)}}[\boldsymbol{g}(Y)]$.

Then, we define $\hat{\boldsymbol{\Lambda}}_{\boldsymbol{f}}$ and $\hat{\boldsymbol{\Lambda}}_{\boldsymbol{g}}$ as the covariance matrices of features on target samples:

$$\hat{\boldsymbol{\Lambda}}_{\boldsymbol{f}} \triangleq \mathbb{E}_{\hat{P}_X^{(T)}}[\tilde{\boldsymbol{f}}(X) \tilde{\boldsymbol{f}}^{\mathrm{T}}(X)], \tag{57}$$

$$\hat{\boldsymbol{\Lambda}}_{\boldsymbol{g}} \triangleq \mathbb{E}_{\hat{P}_Y^{(T)}}[\tilde{\boldsymbol{g}}(Y) \tilde{\boldsymbol{g}}^{\mathrm{T}}(Y)]. \tag{58}$$

In our implementations, all the computations in terms of the underlying distribution are replaced by the corresponding empirical distributions. As proved in Xu & Huang (2020), minimizing $\chi_M^2(\hat{P}_{XY}^{(i)}, \hat{P}_X^{(T)}\hat{Q}_{Y|X}^{(\boldsymbol{f},\boldsymbol{g})})$ $(\hat{Q}_{Y|X}^{(\boldsymbol{f},\boldsymbol{g})}(y|x) \triangleq \hat{P}_Y^{(T)}(y)\left(1 + \boldsymbol{f}^{\mathrm{T}}(x)\boldsymbol{g}(y)\right))$ is equivalent to maximizing

$$H^{(i)}(\boldsymbol{f}, \boldsymbol{g}) \triangleq \mathbb{E}_{\hat{P}_{XY}^{(i)}}[\tilde{\boldsymbol{f}}^{\mathrm{T}}(X)\tilde{\boldsymbol{g}}(Y)] - \frac{1}{2}\operatorname{tr}(\hat{\boldsymbol{\Lambda}}_{\boldsymbol{f}}\hat{\boldsymbol{\Lambda}}_{\boldsymbol{g}}), \tag{59}$$

where $i = T, S$. Then, line 4 in Algorithm 1 can be implemented by

$$L_1(\alpha, \boldsymbol{f}, \boldsymbol{g}) \leftarrow (1 - \alpha)H^{(T)}(\boldsymbol{f}, \boldsymbol{g}) + \alpha H^{(S)}(\boldsymbol{f}, \boldsymbol{g}). \tag{60}$$

Meanwhile, the distance bound $D$ is also estimated from samples. As the simplest way, we let $D = \tilde{D}_1$ ($D$ can actually be adjusted), where

$$\tilde{D}_1^2 = \chi_M^2(P_X^{(T)}Q_{Y|X}^{(\boldsymbol{f}^*, \hat{\boldsymbol{g}}_T)}, P_X^{(T)}Q_{Y|X}^{(\boldsymbol{f}^*, \hat{\boldsymbol{g}}_S)})$$

$$\leftarrow \sum_{y \in \mathcal{Y}} \hat{P}_Y^{(T)}(y) \left(\mathbb{E}_{\hat{P}_{X|Y=y}^{(T)}}[\tilde{\boldsymbol{f}}(X)] - \mathbb{E}_{\hat{P}_{X|Y=y}^{(S)}}[\tilde{\boldsymbol{f}}(X)]\right)^{\mathrm{T}} \hat{\boldsymbol{\Lambda}}_{\boldsymbol{f}}^{-1} \left(\mathbb{E}_{\hat{P}_{X|Y=y}^{(T)}}[\tilde{\boldsymbol{f}}(X)] - \mathbb{E}_{\hat{P}_{X|Y=y}^{(S)}}[\tilde{\boldsymbol{f}}(X)]\right). \tag{61}$$

### E.2  ALGORITHM 2

Here we provide the details of the loss function $L_2(\boldsymbol{f}, \boldsymbol{h}_1, \boldsymbol{h}_2)$ in line 2 of Algorithm 2, the quantity $D$ and the quantity $\tilde{D}_2$ in line 3.

Similarly, we define $\tilde{\boldsymbol{f}}_1(X) \triangleq \boldsymbol{f}(X) - \mathbb{E}_{\hat{P}_X^{(T)}}[\boldsymbol{f}(X)]$, $\tilde{\boldsymbol{f}}_2(X) \triangleq \boldsymbol{f}(X) - \mathbb{E}_{\hat{P}_X^{(S)}}[\boldsymbol{f}(X)]$, $\tilde{\boldsymbol{h}}_1(Y) \triangleq \boldsymbol{h}_1(Y) - \mathbb{E}_{\hat{P}_Y^{(T)}}[\boldsymbol{g}_1(Y)]$, and $\tilde{\boldsymbol{h}}_2(Y) \triangleq \boldsymbol{h}_2(Y) - \mathbb{E}_{\hat{P}_Y^{(S)}}[\boldsymbol{h}_2(Y)]$.

Then the covariance matrices are

$$\hat{\boldsymbol{\Lambda}}_{\boldsymbol{f}_1} \triangleq \mathbb{E}_{\hat{P}_X^{(T)}}[\tilde{\boldsymbol{f}}_1(X)\tilde{\boldsymbol{f}}_1^{\mathrm{T}}(X)], \tag{62}$$

$$\hat{\boldsymbol{\Lambda}}_{\boldsymbol{f}_2} \triangleq \mathbb{E}_{\hat{P}_X^{(S)}}[\tilde{\boldsymbol{f}}_2(X)\tilde{\boldsymbol{f}}_2^{\mathrm{T}}(X)], \tag{63}$$

$$\hat{\boldsymbol{\Lambda}}_{\boldsymbol{g}_1} \triangleq \mathbb{E}_{\hat{P}_Y^{(T)}}[\tilde{\boldsymbol{g}}_1(Y)\tilde{\boldsymbol{g}}_1^{\mathrm{T}}(Y), \tag{64}$$

$$\hat{\boldsymbol{\Lambda}}_{\boldsymbol{g}_2} \triangleq \mathbb{E}_{\hat{P}_Y^{(S)}}[\tilde{\boldsymbol{g}}_2(Y)\tilde{\boldsymbol{g}}_2^{\mathrm{T}}(Y). \tag{65}$$

Still, all the computations in terms of the underlying distribution are replaced by the corresponding empirical distributions. Considering the local approximation that $P_{XY}^{(i)}(x, y) \sim P_X^{(i)}(x)P_Y^{(i)}(y)$, the Hellinger distance loss function $H^2\left(\hat{P}_{XY}^{(T)}, P_X^{(T)}\tilde{Q}_{Y|X}^{(T,\boldsymbol{f},\boldsymbol{h}_1)}\right)$ can be implemented by maximizing

$$\tilde{H}^{(T)}(\boldsymbol{f}, \boldsymbol{h}_1) = \sum_{x \in \mathcal{X}, y \in \mathcal{Y}} \hat{P}_{XY}^{(T)}(x, y)\tilde{\boldsymbol{f}}_1^{\mathrm{T}}(x)\tilde{\boldsymbol{g}}_1(y) - \frac{1}{2}\operatorname{tr}(\hat{\boldsymbol{\Lambda}}_{\boldsymbol{f}_1}\hat{\boldsymbol{\Lambda}}_{\boldsymbol{g}_1}). \tag{66}$$

Similarly,

$$\tilde{H}^{(S)}(\boldsymbol{f}, \boldsymbol{h}_2) = \sum_{x \in \mathcal{X}, y \in \mathcal{Y}} \hat{P}_{XY}^{(S)}(x, y)\tilde{\boldsymbol{f}}_2^{\mathrm{T}}(x)\tilde{\boldsymbol{g}}_2(y) - \frac{1}{2}\operatorname{tr}(\hat{\boldsymbol{\Lambda}}_{\boldsymbol{f}_2}\hat{\boldsymbol{\Lambda}}_{\boldsymbol{g}_2}). \tag{67}$$

Line 2 in Algorithm 2 can be implemented by

$$L_2(\boldsymbol{f}, \boldsymbol{h}_1, \boldsymbol{h}_2) \leftarrow \tilde{H}^{(T)}(\boldsymbol{f}, \boldsymbol{h}_1) + \tilde{H}^{(S)}(\boldsymbol{f}, \boldsymbol{h}_2). \tag{68}$$

We let $D = \tilde{D}_2$ ($D$ can actually be adjusted), where

$$\tilde{D}_2^2 = \frac{1}{2}\sum_{y \in \mathcal{Y}}\left\|\sqrt{P_Y^{(T)}(y)}\boldsymbol{\Lambda}_T^{\frac{1}{2}}\hat{\boldsymbol{h}}_T(y) - \sqrt{P_Y^{(S)}(y)}\boldsymbol{\Lambda}_S^{\frac{1}{2}}\hat{\boldsymbol{h}}_S(y)\right\|^2$$

$$\leftarrow \frac{1}{2}\sum_{y \in \mathcal{Y}}\left\|\sqrt{\hat{P}_Y^{(T)}(y)}\hat{\boldsymbol{\Lambda}}_{\boldsymbol{f}_1}^{\frac{1}{2}}\tilde{\boldsymbol{h}}_1(y) - \sqrt{\hat{P}_Y^{(S)}(y)}\hat{\boldsymbol{\Lambda}}_{\boldsymbol{f}_2}^{\frac{1}{2}}\tilde{\boldsymbol{h}}_2(y)\right\|^2, \tag{69}$$

and

$$\mathbf{\Lambda}_T \leftarrow \hat{\mathbf{\Lambda}}_{\boldsymbol{f}_1}, \tag{70}$$

$$\mathbf{\Lambda}_S \leftarrow \hat{\mathbf{\Lambda}}_{\boldsymbol{f}_2}. \tag{71}$$

Specifically, to compute the value of Bessel functions, we make the following approximations when needed. When $x \ll \nu$, $J_\nu(x) \sim x/v$. When $x \gg \nu$, $J_\nu(x) \sim (2x - \nu)/(2x)$.

## F  INSTRUCTION FOR CODES

We provide code examples in "supplementary_material.zip". In the folder "./cifar10", we provide the code examples on CIFAR-10 dataset for feature extraction (construct_feature_vectors.py), MM-$\chi^2$ algorithm (mmchi2.py), and MM-Hel algorithm (mmhel.py). Folder "./office-31/feature" contains the features of Office-31 dataset. Folder "./office-31/minimax" contains the code examples for Office-31 dataset, including the feature extraction code (feature_extract_4096.py), MM-$\chi^2$ algorithm (oc_atod_vgg16_4096_chi2.py), and MM-Hel algorithm (oc_atod_vgg16_4096_hel.py).

