# OpenReview forum: "Robust Transfer Learning Based on Minimax Principle"
_ICLR.cc/2023/Conference — Submitted to ICLR 2023_

### Official Review · Reviewer_snNr · 2022-10-21

**Confidence:** 4
**Correctness:** 2
**Technical Novelty And Significance:** 2
**Empirical Novelty And Significance:** 2
**Recommendation:** 3

**Clarity, Quality, Novelty And Reproducibility:**

[See the discussions in Strength and Weakness]



**Strength And Weaknesses:**

### Summary

In general, this study investigates an important and challenging topic of estimating joint distribution similarities across heterogeneous tasks in the context of High-dimensional data, the related theoretical analysis is further provided. However, there are several major flaws (such as significance, clarity) in the manuscript that prevent it from being accepted.

### Pros

1. This paper considered robust transfer learning through min-max principle, through distributional robust viewpoint (w.r.t. joint distribution), the key technical difficulty is to measure the joint distribution similarity in both raw data space and representation space.
2. A novel task reweighting based approach is proposed and further validated on standard computer vision benchmark. (even tasks with different label spaces).

### Cons

1. In general there is a significant mismatch between claimed contributions and actual contribution, making this reviewer quite confused.
2. The notations within the paper are seemingly rather unclear, making some parts quite difficult to follow.
3. [Significance] Despite the intensive analysis, this paper still did not theoretically explain why this could ensure a transfer. I mean in terms of **sample complexity**.

### Detailed comments on cons

1. The title “robust transfer” is designed as the main goal. Throughout the whole paper, except the minmax formula, I could not find any sort of clear justification (either in theory or practice) to show the robust transfer is achieved.  Indeed, minmax formula can be regarded as a nature interpretation of robust, while this paper did not clearly present how robustness is learned.

2. In abstract, several key points are highlighted: task similarity, boundness, worst-case expectation. I do have concerns and confusions on all these concepts

- **Task similarity** I have no doubt that task similarity on the joint distribution is quite difficult, while I do think that chi^2 and Hellinger Distance are incorrect metrics in measuring task similarity in high-dimensional and complex data regimes. In experiments, the source and target task could have different distribution support, which clearly violates the assumption in Sec 2, and chi^2 distance will be arbitrarily large in this case. Thus I would think the whole analysis does not exactly match the proposed scenarios. To this end, I would recommend paper [1] through Wasserstein distance (this is more reasonable!) to properly measure the task similarity by considering the parameter and data similarity. In contrast, this paper did not consider (I mean theoretically) the influence of parameters in estimating the task similarity. This gap enables a clear mismatch between the theory and practice in the representation learning setting. Overall, compared with paper [1], this reviewer feels the solutions are not convincing in both theory and practice.

- **boundness** It is quite natural that the source and target should be similar within certain levels of bounded distance. While in practice,the source and target are fixed, this assumption does not make strong sense for me.

- **worst-case expectation** This concept makes me quite confused, the nature of min-max surely ensures the worst-case performance. However, I feel like this paper does not clearly prove/demonstrate the improvement of worst-case loss.

3. Notation/concept clarifications

- In Eq(3), how D is determined?
- Eq(5) seems quite unclear for me because I could not understand why is the exponential form $exp(-d(.,.))$.
- Eq(12) does not make sufficient sense for me, since source samples are much larger than target, and $|X|$ is also quite large, which will enable $\alpha_1 \to 1$.
- Eq(13), $g(y)$ is quite wired, it is essentially the downstreaming predictor?
- Eq(17), I feel like the disconnection between the analyzed theory and proposed loss. Why here the distribution could be decomposed as $P(x)Q_{Y|X}$? How is this related to the analysis?

4. Comments on experiments. I would strongly suggest more analysis rather than numeric accuracy to justify the robust transfer is ensured.

5. About general theoretical analysis in transfer learning. I would strongly suggest the analysis on the sample complexity, which is the key in understanding transfer learning. Unfortunately, the population level loss (and analysis within the paper) could not illustrate why the sample complexity in the target domain is improved.

[1] An Information-Geometric Distance on the Space of Tasks. ICML 2021


**Summary Of The Paper:**

This paper discussed the robust transfer learning through min-max principle. Specifically, this paper adopted chi^2/Hellinger distance to measure the joint distribution. Through min-max formula, this paper focused on the task reweighting approach such that to learn the source target weight, embedding and its downstreaming tasks. This paper is further empirically validated in standard computer vision benchmarks.


**Summary Of The Review:**

This study investigates an important and challenging topic of estimating joint distribution similarities across heterogeneous tasks in the context of high-dimensional and complex data, the related theoretical analysis is further provided. However, there are several major flaws (such as significance, clarity) in the manuscript that prevent it from being accepted.

### Update after rolling discussions

Based on rolling discussions and checking others’ reviews, I am still not convinced. Thus I would keep my current score.

---

> ### Comment · Reviewer_snNr · 2022-11-22
> **Post Rebuttal**
>
> I would appreciate responses by authors. The followings are my additional comments.
>
> > About sample complexity in transfer learning.
>
> I believe this is the key point in theoretically understanding transfer learning. In general, we only have limited samples in the target domain. Thus ERM leads to a high generalization error gap. Transfer learning theories, aim to leverage source information to improve this gap through improved sample complexity. In the theoretical aspect, if derived theories could not be justified when/how sample complexity is improved. It is quite hard to say the theoretical benefits.
>
> > That's why we introduce NN here. A feature extractor maps high-dimensional data to d-dimensional real space. The computation of divergences are made on the real space, which is the same for both the target and source.
>
> Indeed, neural networks could surely be introduced. The problem is it should be also theoretically justified how/when/which neural network is used for improving the transfer learning. For example, I could use a trivial neural network such that it only outputs constant value (e.g, 0), there is no distribution support problem  but it does not make sense, right?
>
> > As for the reviewer recommends Wasserstein distance, we do not want to argue which measurement is better. We believe every measurement can show its advantage according to the setting and purpose.
>
> It should clearly specify why choose metric is better, compared with *previous work* such as paper [1]. We surely can deploy many distribution distance metrics while it should specify and *compare* with previous works to justify the choice.
>
>
>  > The reviewer also says "this paper did not consider (I mean theoretically) the influence of parameters". Let us make it clear that the parameters are functions of the training samples. We train the NN with training samples then we get the parameters in the NN. You may compare the data or compare the parameters. A simultaneous comparison is redundant.
>
> I still could not understand how it is theoretically justified within the paper. A simple example, different network parameters such as LeNet 5 vs. Res 121, which theory could demonstrate the difference of network choice for example?

---

### Official Review · Reviewer_Axde · 2022-10-23

**Confidence:** 5
**Correctness:** 3
**Technical Novelty And Significance:** 3
**Empirical Novelty And Significance:** 3
**Recommendation:** 5

**Clarity, Quality, Novelty And Reproducibility:**

In general, the writing of this work is clear, smooth, and easy to understand. In terms of novelty, I'm not sure whether there has been any other literature studying similar frameworks because this minimax framework is very popular. To me, without appropriate explanations and discussions, the bounded similarity condition is not a good selling point. The use of different f-divergences and the explicit forms of best estimators are novel.

**Strength And Weaknesses:**

Strengths:
1) The minimax framework is quite general and explicit forms of the best estimators are derived;
2) Good numerical performance of the new method;
3) Clear writing;

Weaknesses:
1) The authors claimed that the bounded similarity is a weaker assumption, compared to the usual assumptions in literature. But I think the similarity condition (3) is not well explained. For example, parameter D seems to play a key role in both theory and algorithms. How large should D be to ensure the improvement on the target model compared to the target baseline (say, training models only using target data)? How should users pick D in practice? Your algorithms seem to require the input of D and some other unknown parameters to calculate the best weight, but there are no discussions on this. What are the choices of these unknown parameters in your numerical studies? More clarifications and discussions might be necessary.
2) Some of the mathematical presentations are not very rigorous. For example, before equation (5), what's the meaning of "the probability of the empirical distribution"?  And the following $D$ represents the KL divergence? What does the notation $\mathbb{P}(\hat{P}; P)$ mean? Similar notations appear in other places too. Also, in Theorem 3.1 and some other places, what if $D \rightarrow \infty$ (i.e. the case of negative transfer)? The weight becomes negative?
3) Some model settings and results are a little bit ad-hoc without enough interpretations. For example, the authors considered different underlying models ($(X,Y)$ distributions) when using different f-divergences. Is this just for the simplicity of theoretical analysis? Also, in Section 4, it seems that the neural network (NN) is only used as a dimension reduction tool. If this is the case, can the theoretical results be more general?
4) There are only theories on the best population-level estimators, but no theories on the empirical estimators (which is actually what you used in practice).

Other comments:
1) Is it possible to extend the framework to the case of multiple sources?
2) Can you provide more interpretations of how the results are connected to "robustness"? What is the benefit of using the minimax framework? As you mentioned, the minimax framework considers the worst scenario so it can be more robust to sources with different similarities. But I cannot get this message from your theoretical results and numerical studies.


**Summary Of The Paper:**

The authors provided a general minimax framework to tackle the unknown similarity between the target and source in transfer learning problems. Some specific f-divergences were considered as the similarity measure and population-level best minimax estimators were derived. The best minimax estimators turned out to be a weighted average of the target estimator and source estimator, which is very intuitive. The numerical results demonstrated the effectiveness of the new method.

**Summary Of The Review:**

In general, no matter judging from the problem, the method, or the results, this is an interesting paper. But there are indeed some places that are not well-explained in the paper, as I noted above. These issues need to be fixed and better clarified. It is hard for me to recommend the current version for acceptance. If all the concerns can be properly addressed, I may change my mind, but it's hard to say.

---

### Official Review · Reviewer_Vs9B · 2022-10-25

**Confidence:** 2
**Correctness:** 3
**Technical Novelty And Significance:** 3
**Empirical Novelty And Significance:** 2
**Recommendation:** 6

**Clarity, Quality, Novelty And Reproducibility:**

Clarity: The paper has good clarification and motivation.

Quality: I did not check the proof in the appendix.

Novelty: The paper has its novelty as far as I know.

Reproducibility: I believe the experiments part can be reproduced based on the information provided in the paper.


**Strength And Weaknesses:**

Strength:
1. As far as I know, this is the first paper to consider Chi-square distance and Hellinger distance in transfer learning, and it can efficiently compute the expectations of the population risk.
2. The paper includes the analysis of the continuous data. Also, the paper provides the combining coefficient which is both theoretically optimal and computable from data.

Weakness and Questions:
1. I did not get the sense of robustness mentioned in the title. How to understand the robustness of this work?
2. For equation (13), there is no intuition about using factorization. What is the insight here?
3. The theoretical analysis considers a linear probing setting, while the algorithm uses fine-tuning. Here is a gap between the theory and experiments. I wonder whether we can fix the gap here.
4. It seems the experiment results in Table 3 and Table 4 only gain marginal improvements compared with other state-of-the-art.


**Summary Of The Paper:**

The paper studies transfer learning based on the min-max principle under the Chi-square distance and Hellinger distance. Compared with  KL-divergence distance, the paper uses two distance metrics to fix the difficulty of computing the expectations of the population risk. Under mild assumptions, the paper shows that the optimal estimation is to linearly combine the learning results of the source task and target task, which is consistent with previous work. Based on the theoretical results, the authors proposed an algorithm and evaluated its efficacy on CIFAR-10, Office-31, and Office-Caltech datasets.

**Summary Of The Review:**

Although there are some questions I mentioned in the Weakness part that blocked me, the paper has its novelty and I tend to accept it.

---

### Official Review · Reviewer_cYFo · 2022-10-27

**Confidence:** 4
**Clarity, Quality, Novelty And Reproducibility:** The paper is well organized, but the …
**Correctness:** 3
**Technical Novelty And Significance:** 2
**Empirical Novelty And Significance:** 2
**Recommendation:** 3

**Strength And Weaknesses:**

Strengths:

1.  This paper investigates transfer learning as a minimax problem, which is interesting and novel.

2. The proposed robust transfer learning algorithms are theoretically justified.

3. The paper is well-organized


Weaknesses:

1. The claims made in the paper are not clearly verified. For example, in the abstract, it is said the similarity between the source and target domains is “difficult to be precisely captured”. However, after reading the paper, it is still not clear to me compared to existing quantities, how and why the $\chi^2$-distance and Hellinger distance can “precisely” capture the similarity. Moreover, it is unclear to me why the boundedness assumption is weaker than existing assumptions and notions of similarity (e.g., $\lambda$ and $H$-divergence in [1]). More specifically, I didn't any analysis of the generalization bound of the proposed method. Given that, how can we conclude whether the assumption is \emph{milder } or the proposed estimator can \emph{precisely} capture the similarity?

2. The problem setting studied in this paper is supervised transfer learning, where the label information is available in the target domain. However, it seems that the related work in this field is not reviewed (i.e., theoretical analysis of supervised transfer learning).

3. Another major concern comes from the empirical results, which are quite weak from my aspect. In particular, Office-Caltech and Office-31 are relatively easy tasks compared with more realistic datasets such as Office-Home, VisDA, and mention DomainNet. I believe these (at least the first two) are standard benchmarks for transfer learning and domain adaptation in this field. In addition, the baselines adopted in this paper are not strong enough. For example, in [2], the average accuracies on Office-Caltech and Office-31 are 93% and 89.6%, much higher than the accuracies reported in this paper. Lastly, it is not clear to me how the robustness is empirically verified in the paper


[1] Ben-David, S., Blitzer, J., Crammer, K., Kulesza, A., Pereira, F., & Vaughan, J. W. (2010). A theory of learning from different domains. Machine learning, 79(1), 151-175.

[2] Wang, Q., & Breckon, T. (2020). Unsupervised domain adaptation via structured prediction based selective pseudo-labeling. In Proceedings of the AAAI conference on artificial intelligence.



**Summary Of The Paper:**

This paper studies transfer learning from the aspect of the minimax principle. Accordingly, the strategies of minimizing the worst-case EPR based on $\chi^2$-distance and Hellinger distance are proposed for transfer learning. The effectiveness of the proposed algorithms is supported by empirical results on several benchmark data sets.

**Summary Of The Review:**

While applying the minimax principle to transfer learning is an interesting idea, I encourage the authors to address my concerns and comments before it is ready to publish.

---

### Comment · Area_Chair_iD6y · 2022-11-20
**Update your review**

Dear Reviewers,

Please make sure that your reviews acknowledge authors’ responses and reflect your current evaluation of the paper. This is particularly important if you didn’t directly engage with the authors during the discussion phase (so the authors don’t know if their response changed your evaluation) or if you expressed an intention to update your rating but did not do so.

Cheers,
AC

---

### Decision · Program_Chairs · 2023-01-20

**Decision:**

Reject

**Justification For Why Not Higher Score:**

Lack of experiment and theoretical results to support the claims. The reviewers converge to the negative side.

**Justification For Why Not Lower Score:**

N/A

**Metareview: Summary, Strengths And Weaknesses:**

This paper studies transfer learning based on the minimax principle that quantifies the similarity between the source and the target domain with Chi-square and Hellinger distances. The authors try to minimize the worst-case expected population risk of transfer learning and build a robust transfer learning approach.

The reviewers noted that, although the paper is well written, the novelty of introducing the minimax principle into transfer learning. And the reviewers point out their concerns:
a) The authors do not clearly present their insights on the robust transfer, boundness, and gap between theories and experiments.
b) The experiments should be strengthened to support their claims
The authors clarified some points in their response, but the paper would still require some more modifications to be ready for publication.


**Summary Of Ac-Reviewer Meeting:**

The reviewers showed this work support this line of research, a minimax principle introduced into transfer learning study.
The discussions point out the main concerns of reviewers. The authors also made active rebuttals and clarified their claims in robust transfer learning. There are many reviewers who still are still expressing concerns.
Overall, this is an important line of research, and I very much encourage the authors to improve in expressing their insights and submit to a different venue.